# Creating the Urban Farmer’s Almanac with Citizen Science Data

**DOI:** 10.3390/insects10090294

**Published:** 2019-09-11

**Authors:** Kathleen L. Prudic, J. Keaton Wilson, Michelle C. Toshack, Katharine L. Gerst, Alyssa Rosemartin, Theresa M. Crimmins, Jeffrey C. Oliver

**Affiliations:** 1School of Natural Resources and the Environment, University of Arizona, Tucson, AZ 85721, USA; 2Volunteer Experience, Adventure Scientists, Bozeman, MT 59771, USA; 3National Coordinating Office, USA National Phenology Network, Tucson, AZ 85721, USA; 4Office of Digital Innovation and Stewardship, Libraries, University of Arizona, Tucson, AZ 85721, USA

**Keywords:** eButterfly, iNaturalist, insect management, urban insect diversity, USA National Phenology Network, Nature’s Notebook

## Abstract

Agriculture has long been a part of the urban landscape, from gardens to small scale farms. In recent decades, interest in producing food in cities has grown dramatically, with an estimated 30% of the global urban population engaged in some form of food production. Identifying and managing the insect biodiversity found on city farms is a complex task often requiring years of study and specialization, especially in urban landscapes which have a complicated tapestry of fragmentation, diversity, pollution, and introduced species. Supporting urban growers with relevant data informs insect management decision-making for both growers and their neighbors, yet this information can be difficult to come by. In this study, we introduced several web-based citizen science programs that can connect growers with useful data products and people to help with the who, what, where, and when of urban insects. Combining the power of citizen science volunteers with the efforts of urban farmers can result in a clearer picture of the diversity and ecosystem services in play, limited insecticide use, and enhanced non-chemical alternatives. Connecting urban farming practices with citizen science programs also demonstrates the ecosystem value of urban agriculture and engages more citizens with the topics of food production, security, and justice in their communities.

## 1. Introduction

Urbanization is a major driver of land use change worldwide [1]. By 2030, more than 60% of the global population will live in urban areas [2] and transform many suburban and rural agricultural systems into urban environments [3]. Urban agriculture is defined as the production of crop and livestock goods within cities and towns [4], generally integrated into the local urban economic and ecological system [5]. It has emerged as a tool to address complex social issues, such as environmental justice, food security, and income inequity [6]. Urban agriculture provides resources and shelter for urban animals beyond humans, enhances biodiversity, and improves ecosystem services, making cities more resilient and resistant to environmental change [7]. Cities are primarily viewed in terms of their political value (i.e., where the voters are) rather than for their ecological value (i.e., where food, shelter, water, and mates are) [8]. Urban environments are predominantly viewed by both scientists and the general public as biodiversity deserts responsible for high rates of extinction [9,10] and reduced abundance, especially of native species [11,12]. Yet, in many urban areas, the patchwork of formal and informal green spaces provides viable and important habitat for a diverse selection of plants and animals [13,14]. Managed green spaces, such as farms and gardens, provide space and resources are critical for the long-term preservation of urban wildlife, such as insects and birds, and enhance local ecosystem services, improving water, air, and soil quality [15,16]. Urban farms and gardens also present important opportunities to connect urban dwellers with nature and grow their appreciation for their non-human animal and plant neighbors [17].

Insects play many roles in agriculture systems; they are categorized primarily as pest, benficial, and pollinator. Pests are often defined as insects that harm yields and/or the quality of crops. Beneficials include predators, parasitoids, and scavengers which indirectly benefit crops by consuming pests or reducing waste. Pollinators fertilize crops by moving pollen from one flower to another incidentally, as these insects collect pollen and nectar for their own consumption. A single insect species may fill multiple categories, and designations may change throughout various life stages. For example, a hawk moth may be a valuable pollinator as an adult, but a voracious pest consuming a vast amount of crop leaves as a larva [18]. Identifying and monitoring the insect community in agroecosystems is a critical component to success in farming, especially urban farming where farms are hotspots for insect biodiversity [8].

Agroecosystems, including urban farms, benefit from insect diversity and phenology data to inform management decisions. By developing shared knowledge of the presence of pests, urban growers can make informed decisions and assess the efficacy of the treatments they apply. Correct insect identification is critical for determining which control actions, if any, should be taken to minimize damage from insect pests. Phenological information indicating when a pest is anticipated to be found in an important life stage can further enhance management planning [19]. This phenological perspective also supports management decisions such as applying a pesticide when it is least likely to impact a pollinator [20,21] or planting flowering plant species that bloom during a gap in blooming, where pollinators have reduced resources [20,22]. Data and tools offered by citizen science programs can support urban agroecosystems in all of these ways.

## 2. Urban Insect Management Presents Unique Challenges and Opportunities

Urban farming presents both challenges and opportunities related to the environmental setting and the demands of meeting multiple social and educational goals. The challenges include difficulty accessing land, small and fragmented plots close to residences and businesses, soil contamination, and insecure land tenure [23,24]. Insect management is challenging in urban settings from a social standpoint, given the varying values and expertise levels of neighbors or community gardeners. In addition, controlling urban insect pests requires limited pesticide use to ensure the health of humans and animals. Excessive application of pesticides can degrade water, air, and soil quality, create pesticide resistant insect populations, and be economically costly to the grower [7] and many broad-spectrum management interventions for pests may have undesired non-target effects on pollinators and beneficials [25]. For example, organophosphate application increases slug pest abundance and crop loss because of decline in predaceous beetles [25]. Finally, given the reduced availability of native plants, undergrowth, and connectivity among sites in urban landscapes, beneficial insect populations may be harder to attract and sustain in urban areas, relative to rural agroecosystems [26]. The opportunities include community building, awareness of food and agriculture, and access to healthy foods [27,28]. Similarly, urban agriculture’s popularity stems from its many benefits for the individual, community, and city as a whole [6] such as improved physical activity and mental health [29], nutrition [30], community engagement [31], and job training [32]. Urban agriculture has expanded by >30% in the past 30 years, especially in under-served communities [33]. Urban agriculture can be productive, providing an estimated 15%–20% of the global food supply [34,35], and cities can provide good infrastructure, access to labor, and low transport costs for local food distribution [34].

Although public and scientific interest in urban agriculture has grown dramatically in the past two decades, there are still significant challenges for integrating urban farming into the complex agriculture support system in the United States and beyond [13]. The United States Cooperative Extension System was developed when most of the population lived and farmed in rural environments [36] and agriculture research has and continues to be done primarily on rural farms [7]. Recent efforts by Cooperative Extension have incorporated urban engagement and farming with success [37]; however, many of these efforts have focused on nutrition, food literacy, and youth leadership training (e.g., 4-H), all of which are important community-driven issues for urban stakeholders, but not urban farming pest management best practices [38]. Other countries with different systems of agriculture support (e.g., Canada [39], Tanzania [40]) also struggle with the pace of urban farm implementation without a concomitant investment in urban agriculture insect management research and best practices. However complicated, urban farms provide important green space and food security to the benefit of both humans [27] and insects [8].

Insect management resources and knowledge of best practices are fewer and less developed for urban growers than those for rural growers [13]. A critical tool for reducing the impact of destructive insects in agricultural systems is integrated pest management (IPM). IPM is a decision framework for the selection and use of pest control tactics, coordinated into an overall management strategy based on cost/benefit analyses that take into account the interests of and impacts on producers, society, and the environment [41]. The implementation and adaptation of IPM in the socio-ecological context of urban farming could provide a powerful framework for leveraging the strengths and mitigating the challenges. This approach reduces the frequency and intensity of pest infestation by eliminating disruptive pest control methods and enhancing ecosystem services that contribute to ecological resilience. In an IPM framework, agriculture systems are managed as living systems [42]. Essential to this framework is documenting where and when insect pests, beneficials, and pollinators are present on the farm and in the surrounding area [42]; such data are increasingly available through a number of citizen science resources.

## 3. Urban Insect Management Can Be Facilitated by Citizen Science

Citizen science relies on the participation of non-professionals in the practices of science, from study design to data collection [43]. Most citizen science data are collected in urban areas (e.g., [44,45]), and both urban farms and citizen science are conduits for community building and civic participation. Additionally, citizen science aligns with many social media outlets to promote a farm with a variety of stakeholders and potential customers. A number of powerful tools and platforms exist to build the connections and collect data required for successful agroecosystem management in the urban socio-ecological context. While there are many citizen science practices, we focused on web-based programs which focus on biodiversity and serve to identify species, store data, and synthesize patterns of diversity.

In this section, we highlighted three web-based tools providing measures of insect diversity and phenology: eButterfly [46], iNaturalist [47], and Nature’s Notebook [48] (Figure 1). eButterfly is designed for butterfly enthusiasts who photograph and checklist butterflies for recreation and it covers North American species. iNaturalist is designed for biodiversity enthusiasts, those who photograph and observe all organisms, including insects, across the globe. Nature’s Notebook, the phenology observing system operated by the USA National Phenology Network, is designed for backyard enthusiasts who wish to track the seasonality of organisms, such as when they are emerging, leafing out, or flowering in the United States. Nature’s Notebook has a tailored list of plants and animals, particularly amenable for observing phenological changes. Using these applications and their various features can be facilitated by online and face to face trainings.

Urban growers can use these platforms to (1) support insect identification, (2) see what insect species are in the surrounding area, (3) connect with local insect enthusiasts and experts, (4) store insect data from the farm in one location, (5) contribute to the shared knowledge about urban insects, (6) predict when insects will be present and abundant, relative to plant development, to guide management decisions, (7) predict when insect pests will be most vulnerable to treatment, and (8) demonstrate the value of urban farms to insect habitat (Figure 2). The exact approach will likely vary by farm location, crops, and mission; however, we feel the greatest potential strength of these programs is to help growers to increase the presence of insect pollinators and beneficials through habitat on the farm and in the community while simultaneously decreasing insect pests.

### 3.1. Citizen Science Provides and Organizes Identifications of Insects in the Farm, Neighborhood, and City

Correctly identifying insect species can be an overwhelming process for the uninitiated, as insects are the most diverse group of animals on the planet, with over 800,000 described species. Traditionally, resources to support identifying insects in the United States included Cooperative Extension and entomological collections. Cooperative Extension provides insect-related expertise mainly to rural growers, though it has recently expanded into urban farming (e.g., [7]). Regional entomology collections located at universities, museums, and agriculture stations also offer identification opportunities [49]. Neither of these resources can provide instant identification feedback due to the multiple other commitments on the institution staff’s time, such as research, instruction, and/or outreach [49]. Furthermore, these resources may not hold expertise relevant to urban landscapes [36].

The human-computer networks offered by web-based citizen science projects enhances opportunities for urban growers to identify insect species. The iNaturalist web-platform and smartphone application [47] and the related Seek smartphone application [50] are the most versatile digital tools available for this purpose. Both of these applications employ artificial intelligence algorithms to identify a plant or animal from a photo. A grower can upload a photograph to their iNaturalist account and suggestions for the species are provided. In the case of the Seek application, a grower can simply point their smartphone camera at the plant or animal and receive a suggested identification. The algorithm originally developed for classifying organisms on iNaturalist is highly accurate, offering the correct identification among the top 5 suggestions between 87.5% and 88.2% of the time [51]. Complementing machine-learning algorithm identifications, iNaturalist relies on the large citizen science community, including trained experts, to provide identification recommendations linking to other observations and photographs. Half of all records of unidentified species that are uploaded and crowdsourced are identified in less than two days. eButterfly identifies butterflies through a slightly different approach of human-computer interaction [46]. In the eButterfly app, a series of filters based on current species distribution maps are coupled with regional experts to flag whether a species listed in a checklist is expected at a specific location. Regional experts work with citizen scientists to identify unexpected species from photographs and descriptions. While species identification is not a primary feature of the Nature’s Notebook smartphone application, materials to support species identification are offered on the Nature’s Notebook website.

In addition to serving as data collection and storage systems, these citizen science programs also provide several easy-to-use dashboards for managing and visualizing data. Such visual representations provide an accessible means of gauging insect and plant phenology at a local scale. iNaturalist offers the ability to store and filter all insect data by location and date [47]. These data can be displayed in a variety of ways, providing information important to management such as emergence time, diversity, and abundance. Data from a single farm can be aggregated with other local observations to form a more complete picture of the surrounding area. As in iNaturalist, data housed in eButterfly can be filtered by location and date [46]. eButterfly data are presence-absence data of butterfly species (pollinators and pests) while iNaturalist data are presence-only data of all insects (beneficials, pollinators, and pests), providing different kinds of data for different kinds of information and decision making [46]. For both iNaturalist and eButterfly, a grower can have their own account to record observation data and photos across years at their farm and in the community. Nature’s Notebook displays data on a focal insect or plant species and is particularly good for documenting an organism’s life cycle stage status over the course of a season.

### 3.2. Citizen Science Provides Information on When Insects Will Be Active and Abundant

Phenology, or the seasonality of organisms, has been long viewed as a tool to understand the best time to plant and harvest crops, as well as to anticipate when to manage for insect pests and facilitate pollinator and beneficial insect health. In many systems, environmental conditions such as the accumulation of heat units (i.e., growing degree days), can be utilized to predict when species of interest will undergo phenological transitions, such as the hatching of caterpillars or the emergence of adult leaf beetles (e.g., [52]). Resources such as the USA-NPN Pheno Forecasts offer daily maps and forecasts up to six days in advance [53], which can be used by urban growers to anticipate the activity of insects (Figure 3). iNaturalist and eButterfly provides seasonality estimates based on community-contributed observations and allows users to indicate the life stage of organisms they observe, further guiding proactive management practices. Additionally, participants in the Nature’s Notebook program can provide their own observations of insect activity at their location by participating in the Pest Patrol campaign [54] or Nectar Connector campaign [55]—allowing for the verification and improvement of these predictive models. Ultimately, more data on a greater diversity of insect taxa will lead to more accurate models that can account for variations in climate and geography and lead to improved decision-making for urban growers.

By harnessing the power of citizen science programs, growers can be empowered within their communities to better establish the spatial and temporal patterns of insects across urban and suburban landscapes at scales not previously possible by scientific researchers. By participating in citizen science programs, such as those described above, individuals join a collective effort to track both beneficial and harmful insects which in turn helps inform best practices for enhancing growing environments. Early work indicates that insect pest species tend to be more abundant in urban areas due to warmer conditions [56] where some beneficial insects, such as wild bees, decline in urban areas [57]. Likewise, species composition tends to shift along urban to rural gradients (e.g., [58,59]). In addition, urban environments tend to have earlier and longer growing seasons, impacting the timing and magnitude of the interactions between plants, pollinators, and herbivores [60]. Such patterns offer glimpses of predicted impacts of climate change to communities and ecosystems [61], with more thoughtful urban landscape design as an approach to buffer against some of these changes [62,63].

## 4. Digital Collaboration Creates Useful Information for Everyone

The three citizen science programs described here amplify the collaborative nature of urban farming using online resources. Urban growers often engage with a large community to become sustainable economic enterprises and to fulfill other missions, such as food literacy education [30] and job training [32]. eButterfly, iNaturalist, and Nature’s Notebook allow urban growers to expand their community to a larger audience, connect with experts in insect identification and management, evaluate how local changes fit into a larger regional context, and help sustainably manage urban green spaces as not only viable businesses but also as centers for biodiversity and green space. These citizen science programs are available in a variety of languages (e.g., English, French, Spanish), depending on the web-platform and smartphone application. In general, iNaturalist has the most languages with over ten.

While most citizen science programs were originally designed to operate independently, there is increasing integration among the various systems. Part of this integration is due to advances in application programming interfaces (APIs) that afford easier sharing of data. For example, iNaturalist sends “research grade” observations to the Global Biodiversity Information Facility [64] on a weekly basis. Third-party citizen programs are using multiple platforms for monitoring diversity: the Appalachian Mountain Club employs both iNaturalist and Nature’s Notebook platforms for data collection. Coordinated efforts, including scavenger hunts and “BioBlitzes” improve biodiversity knowledge of urban greenspaces, and urban farms could be easily interwoven into this network through face-to-face and social media connections.

High-quality citizen science programs rely on feedback from participants; this includes feedback on the web-based platforms discussed here. Urban growers should not hesitate to contact citizen science directors to suggest new features for the web-platforms and smartphone applications. Indeed, insect diversity dynamics unique to urban agroecosystems may necessitate qualitative or quantitative changes to the way data are collected. Sometimes, these requests are very easy for web designers to incorporate into their updates or new versions, while more complex changes may require more time or be infeasible. However, these conversations improve the products and fosters a sense of community between scientists, programmers and participants.

## 5. Conclusions and Recommendations

Recent trends show an increase of agricultural efforts within urban areas, in both developed and developing nations [13]. Urban farming has significant potential to enhance local communities in a variety of ways beyond food production [32]. Integrating urban agroecosystems into the local greenspace matrix can support beneficial insect species, providing opportunities for people to appreciate and connect with insects. Managing urban farms to increase insect pollinators and beneficials while controlling the potential damage caused by insect pests will benefit through engagement with local biodiversity enthusiasts. Capitalizing on citizen science efforts will greatly improve safe and effective insect management practices on urban farms. As part of an urban integrated pest management approach, we recommend that growers incorporate citizen science web-platforms such as eButterfly, iNaturalist, and Nature’s Notebook into their farming approach. These tools provide growers with a digital toolkit for promoting pollinators and beneficial insects, decreasing pest species, connecting with entomology experts, and marking their farms as urban biodiversity hotspots.

## Figures and Tables

**Figure 1 insects-10-00294-f001:**
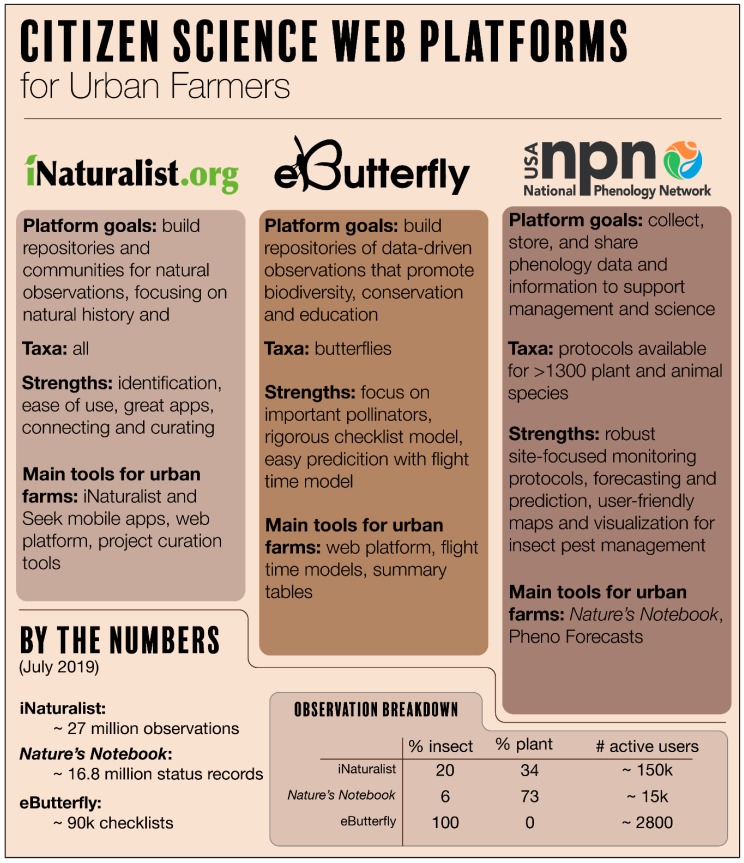
Comparison of iNaturalist, eButterfly, and USA National Phenology Network. Three citizen science platforms offer growers various tools, data, information, and social networks to improve the management of their farm insects, including beneficials, pests, and pollinators.

**Figure 2 insects-10-00294-f002:**
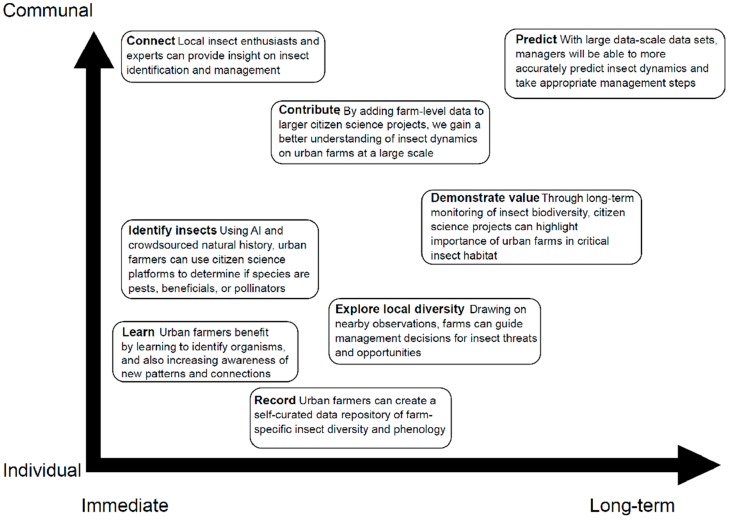
Value schematic of citizen science web-platforms for urban growers. Collaborative citizen science programs add value to urban farm mission and management by offering connection to experts, customers, and community members. Here, we outlined the main ways that urban farms can use citizen science platforms, across two axes: immediate and long-term usage (horizontal) and individual and communal actions among users (vertical).

**Figure 3 insects-10-00294-f003:**
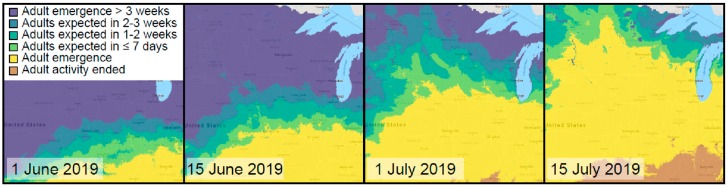
Phenological forecasts. The National Phenology Network’s Pheno Forecast tool (www.usanpn.org/data/forecasts) shows predicted emergence dates for common agricultural pests. This figure shows single-day forecasts for the apple maggot fly, *Rhagoletis pomonella* (Walsh) in the upper Great Lakes region of the United States of America. Source: USA National Phenology Network, www.usanpn.org.

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
