# Peer review of "Creating the Urban Farmer’s Almanac with Citizen Science Data"

_insects, 2019, doi:10.3390/insects10090294_

Round 1
Reviewer 1 Report
A good paper, very relevant to the present time!
A few suggestions are marked in the review copy.
Most important suggestion would be to make the introduction and the subsequent two sections a little shorter and more concise, so that readers can quickly arrive at the essence of the article: which is the three cool tools you are describing.

Author Response
Many thanks for the thoughtful and constructive review. We appreciate your time, expertise, and commitment to making this a better manuscript.
In regards to the references, we included author names for the review process so that the reviewers didn’t spend as much time flipping back and forth to the reference section. You are absolutely correct that this journal uses numbers only. This change along with the other suggested reference changes have been made in this revision. We struggled with the right amount of references, so many more publications these days and less dissemination costs (e.g. online journals) led us to keeping the references. Also we looked at what other folks where doing in the special issue and the average was 74 citations. For better or worse, we came in below this average at 64. We wanted to give the broad audience ideas on where to gather further information and give credit where credit is due.
The introduction was shortened per the suggestion and now has 3 instead of 4 paragraphs. This reduction also contributed strongly to the fewer number of references.
Point by point responses have been shared here
https://drive.google.com/file/d/1rmFJzk0MXJeGhNY1MZE0YT3HKef5Fswo/view?usp=sharing
Again many thanks for your thoughts and constructive feedback. It is appreciated.
Reviewer 2 Report
Dear Authors,
please consider the suggestions I put in to reviewed and UL .pdf file.
Thank you

Author Response
Many thanks for your review. We appreciate your time and expertise.
We found many of the suggestions to be misunderstandings across disciplines and scales of inquiry. We appreciate there are different perspectives and vocabulary across disciplines which need cooperation and discussion to resolve. We expect this type of cooperation during the review process.
We addressed specific comments regarding communication clarity such as changing integrative to integrated. Full details and point by point responses can be viewed https://drive.google.com/file/d/1-MlanbzwDbMW0l7YPGCCTsesdzeJP96K/view?usp=sharing
The reviewer’s concerns we did not specifically address can be summarized as follows.
Ecological vocabulary: There seems to be a general misunderstanding of the ecological terms used in this manuscript such as biodiversity, abundance, native, and phenology. We use the ecological meaning of the words and definitions are provided in instances where there may be disciplinary confusion. Our usage is in line with our supporting references and the broader field of ecology while linking to pest management specifically IPM.
How models scale: Thinking across scales is difficult in any discipline and has computational limitations. Yet, these tools are becoming more universal across disciplines and the previous computational challenges are lowering. In this manuscript, we think about large scale environmental factors such as growing degree days as factors in our predictive model of when an insect would be emerging or arriving in an area much like a storm. The concerns raised by the reviewer regarding growing degree days at the urban farm would be estimated in the models across large spatial scales so particular site fluctuations in temperature and precipitation are not relevant to the predictive models persay.
Pesticide usage to control insects: We all agree pesticide usage is a powerful tool in any pest management toolbox. Based on the work of others, it seems urban farms will have a very different pesticide toolkit based on regulations, farm branding, cost, and effectiveness. We do not feel all pest control treatments will be chemical pesticides, some may be pheromone, some may be mechanical, some may even be sound. We expect there to be innovation and creative thinking on the part of urban farmers in line with their financial resources and their ethos moving forward. This is not an article about specific pest management strategies, but rather a guide to how the greater community can provide useful information on where insects are, what they are doing, and when they are abundant, all in order to enhance urban pest management approaches, especially as new practices are being developed in real time.
Technology in agriculture: The world is changing fast and technology is making strides in helping farmers, including non-traditional farmers. The technology advances we describe here are always improving. Of particular concern for Reviewer 2 is the artificial intelligence driven computer vision identification. We recommend the references we cite for a more complete discussion of how this works. In short, this technology is evolving rapidly and these algorithms are improving at an astounding rate, particularly in areas where experts are engaging with computer scientists. Some species are difficult to identify and may never be applicable for AI; however, many will be identifiable in ways that are different from how we identify species now. These technology tools are valuable, especially when farmers do not have immediate access to a local taxonomic expert. The language in this section is guarded while still providing room for future improvements. We advocate for an inclusive perspective on species identification, which is probabilistic and can account for error, whether that is human- or computer-based.
Experts are not involved with citizen science projects: Several comments by Reviewer 2 suggest a bias against citizen science programs providing valuable data and worthwhile scientific enterprises. A growing body of literature demonstrates the contrary (see https://esajournals.onlinelibrary.wiley.com/doi/full/10.1002/fee.1436). Indeed, experts are flocking to these data because it allows exploration of different questions at different scales. No data are without flaws and just because a non-expert collected it, doesn’t mean it is less valuable. All data should have quality assurance and quality control measures in place, regardless of the source. Furthermore, across the three web platforms listed in this article, 10-40% of data contributors have advanced degrees, so a non-trivial quantity of the data are being contributed by experts.
Reviewer 3 Report
Please see the attached document to review questions and comments.

Author Response
Many thanks for the thoughtful and constructive review. We appreciate your time, expertise, and commitment to making this a better manuscript.
Point by point changes can be found here https://docs.google.com/document/d/1-iJuiI-HPa0OkKSBN9_-3S1QMpSIWXnWeR_rukWNaTM/edit?usp=sharing
All suggested word changes have been made in the manuscript.
Figure 1 has been updated per your suggestion.
Figure 3 suggestion has been moved to the manuscript text providing information on the online and face to face training offered.
Citation for lines 255-257 has been added.
Text now has more information on the various languages these apps are offered in. Thanks for the suggestion.
Again many thanks for your thoughts and constructive feedback. It is appreciated.
Round 2
Reviewer 2 Report
Please delete "singly" (page 3, line 15th); you have any integration if you use a single control means, an not a "tactic."
A need to correctly define IPM.

Author Response
Many thanks for your comments and concerns.
We have altered the text per your request regarding the definition of IPM highlighted in green
This definition is taken directly from Kogan 1998 the cited reference (definition section) The Kogan 1998 paper can be found here if you are interested https://www.annualreviews.org/doi/full/10.1146/annurev.ento.43.1.243